# Differential Proteomic Analysis of Low-Dose Chronic Paralytic Shellfish Poisoning

**DOI:** 10.3390/md22030108

**Published:** 2024-02-26

**Authors:** Xiujie Liu, Fuli Wang, Huilan Yu, Changcai Liu, Junmei Xia, Yangde Ma, Bo Chen, Shilei Liu

**Affiliations:** State Key Laboratory of NBC Protection for Civilian, Beijing 102205, China

**Keywords:** paralytic shellfish poisoning, differential proteomic, synaptic transmission, voltage-gated ion channel

## Abstract

Shellfish poisoning is a common food poisoning. To comprehensively characterize proteome changes in the whole brain due to shellfish poisoning, Tandem mass tag (TMT)-based differential proteomic analysis was performed with a low-dose chronic shellfish poisoning model in mice. A total of 6798 proteins were confidently identified, among which 123 proteins showed significant changes (fold changes of >1.2 or <0.83, *p* < 0.05). In positive regulation of synaptic transmission, proteins assigned to a presynaptic membrane (e.g., Grik2) and synaptic transmission (e.g., Fmr1) changed. In addition, altered proteins in nervous system development were observed, suggesting that mice suffered nerve damage due to the nervous system being activated. Ion transport in model mice was demonstrated by a decrease in key enzymes (e.g., Kcnj11) in voltage-gated ion channel activity and solute carrier family (e.g., Slc38a3). Meanwhile, alterations in transferase activity proteins were observed. In conclusion, these modifications observed in brain proteins between the model and control mice provide valuable insights into understanding the functional mechanisms underlying shellfish poisoning.

## 1. Introduction

In today’s society, global food safety is facing numerous challenges, and the incidence of food safety incidents is increasing day by day. The frequent occurrences of harmful algae blooms (HAB) lead to the enrichment of shellfish toxins in seafood, which brings great security risks to human beings [1]. These toxins are released and accumulate through gill absorption in bivalves, such as mussels and oysters. Shellfish toxins exist in a bound state but quickly dissociate upon entering the human body, resulting in toxicity. There are many types of shellfish toxins, including paralytic shellfish toxins (PSTs), diarrhetic shellfish toxins (DSTs), neurotoxic shellfish toxins (NSTs) and amnesic shellfish toxins (ASTs). The widespread distribution, potent toxicity, and significant harm caused by PSTs have garnered extensive attention. Globally, there are over 2000 reported cases of PST poisoning annually, resulting in a mortality rate of 15% [2]. The first reported case of death due to shellfish poisoning occurred in Colombia, South America, in 1973 [3]. Since then, many cases of shellfish poisoning have been reported in various countries. In 1987, PST poisoning occurred in Guatemala, causing 187 people to be poisoned [4]. In China, most cases primarily occur in Fujian Province, and also other regions [5]. In 1988, the first case of PST poisoning caused by red tide was reported in Fujian Province, 136 people were poisoned, 59 were seriously ill and 1 died [6]. Subsequently, poisoning cases have been frequent, including one in Zhangzhou, Fujian province, during 2017, which poisoned 164 people [7].

PSTs are highly specific to the nervous system of animals. Consumption of aquatic products with PST content as high as 80 μg/100 g can lead to numbness, accompanied by headache, convulsion, paralysis, shock and other neurotoxic symptoms. In severe cases, it can even cause respiratory paralysis [8]. Previous studies have focused on the long-term degenerative effects of STX on the central nervous system. Li et al. explored the cognitive impairment and hippocampal damages after 6 months’ exposure of low-dose STX with C57BL/6NJ mice, and the result highlights the role of Ppp1C and YAP1 cytoplasmic retention in chronic low-dose STX intoxication [9]. Sun et al. explored the effects of long-term low dose saxitoxin exposure on nerve damage in C57BL/6 mice via hippocampal proteomics analysis [10]. The ‘no observed adverse effect levels’ (NOAELs) and ‘lowest observed adverse effect levels’ (LOAELs) are used to determine the level of exposure at which adverse health effects begin to occur. PSTs have a LOAEL of 1.5 μg STX/kg body weight and a NOAEL of 0.5 μg STX/kg body weight based on exposure assessments in mice. If PST intake is NOAEL, the numbness usually subsides after 24 h and the body recovers after 48 h. Patients with this type of poisoning do not produce any protective antibodies, and the next time they re-ingest PSTs, the patient will continue to experience the same intoxication [11]. Currently, more than 50 types of PSTs have been identified with carbamate toxoid, saxitoxin (STX), being the most prevalent, accounting for over 85% [12]. As potent membranous neurotoxins, PSTs exhibit high affinity dysfunction towards voltage sensitive sodium ion channels (VSSC) present in nerve cells. PSTs typically bind to cell membrane ions, including an imbalance in the flow of normal ions across the cell membrane, and resulting in abnormal membrane potential, leading to symptoms such as dizziness, shock, and other neurotoxic effects. Symptoms of STX poisoning are dose-dependent and can occur within 10 min after oral administration of 400 g shellfish contaminated with PSTs at the current regulatory limit (800 µg STX.2HCl eq/kg shellfish flesh) [13]. Although acute toxicity studies on PSTs have been extensively conducted, the chronic low-dose toxic effects of PSTs in humans and other animals remain unclear.

The purpose of this study was to address whether chronic shellfish poisoning leads to potential neurotoxicity in the mammalian system. In particular, the analysis of proteome changes in a low-dose chronic shellfish poisoning model has rarely been reported. A better, more comprehensive understanding of the proteomic level of chronic shellfish poisoning is therefore crucial and valuable. Only Sun et al. have explored the effects of long-term low dose saxitoxin exposure on nerve damage in C57BL/6 mice, via hippocampal proteomics analysis [10]. In this study, 5-week-old mice were exposed to low asymptomatic doses of PSTs via drinking water for 10 weeks to observe behavioral changes in the entire brain. TMT combined with LC–MS analysis was performed to investigate differentially expressed proteins in the entire brain of shellfish poisoning model Kunming mice. The protein profile changes in response to shellfish poisoning were also determined and identified. The results of this study will be helpful to the prevention and treatment of shellfish poisoning.

## 2. Results

### 2.1. Shellfish Poisoning Modeling and Evaluation

During the experiment, no death occurred in the control group and the shellfish poisoning model group. Figure 1 demonstrates a significant decrease in the weight of mice in the model group compared to the control group from the second week to the fifth week. The weight discrepancy in the second week amounted to 7.05 g, in the third week 10.07 g, in the fourth week 10.52 g and in the fifth week 7.46 g. From the sixth week to the tenth week, there was no significant difference in body weight between model group and control group.

### 2.2. PST Impaired Cognitive Performance of Mice

Experiments commenced on the first day of administration in the last week, and a place navigation test (PNT) was conducted on each group of mice for a total of 5 days. As the training duration increased, both groups exhibited a gradual reduction in the average escape latency period (Figure 2A). The spatial learning ability of mice in each group was improved during the 5-day training period. Compared to the control group, the model group showed significantly longer probe time (*p* < 0.05). During the trial phase, after removing the platform, we recorded the frequency with which mice crossed over to where it had been located (Figure 2B–D). We assessed whether there were differences in training time between different doses of PST exposure by recording search time within correct quadrants. Model mice took considerably longer to find the target quadrant, crossed the platform less frequently, and spent less time in the correct quadrant than that in control mice (*p* < 0.05). Taken together, our data suggested that PSTs impaired the cognitive performance of mice.

### 2.3. TMT Analysis of Differentially Expressed Proteins

To uncover differentially expressed proteins in PST-induced neurotoxicity, brains were isolated from control and model group and analyzed using the TMT label with LC–MS. A total of 6798 proteins were identified and quantified against the musculus Database, whereas 123 altered proteins changed in the model group compared to controls (fold changes of >1.2 or <0.83, *p* < 0.05 as threshold). A total of 87 up-regulated proteins and 36 down-regulated proteins were found according to fold changes (Figure 3, Appendix A). Correlation of the protein intensity is shown among four biological replicates of control and shellfish poisoning model groups (Figure 4). The results show that there was obvious clustering between control and model groups, and the quantitative proteomics data sufficiently explains variance between control and model groups. Hierarchical clustering was generated to display the differential proteins across the two groups (Figure 5). Consequently, the altered proteins were clustered into classes based on distance, regarding the up-regulated proteins. Different color blocks represent the relative expression of corresponding proteins and distinct group patterns of differentially expressed proteins.

The differential proteins in Gene Ontology (GO) were analyzed using DAVID. Subsequently, the cellular component (CC), molecular function (MF) and biological process (BP) of all identified proteins, all altered proteins, up-regulated proteins and down-regulated proteins were annotated separately. In terms of CC annotation, there was a noticeable increase in the proportion of membrane-associated proteins among all altered proteins compared to all identified proteins. Conversely, the percentage of intracellular region-assigned proteins (cytoplasm and cytosol) decreased significantly in all altered proteins when compared to all identified ones (Figure 6). Up-regulated proteins exhibited significant clustering within polysomes during MF annotation (Figure 7A). In MF annotation, compared to all identified proteins, the percentage of altered proteins increased in identical protein binding and protein homodimerization activity; these proteins decreased in protein binding (Figure 7B). Notably, significant differences were observed between up-regulated and down-regulated protein clusters across CC, MF or BP during annotation analysis (Figure 7). In BP annotation, the percentage of proteins assigned to the ion transport in all altered proteins increased obviously compared to all identified proteins (Figure 6A). Furthermore, based on GO annotation analysis, clusters of nervous system development, positive regulation of synaptic transmission, ion transport and transferase activity were significantly regulated in model group compared to those in the control group.

#### 2.3.1. Nervous System Development

The expression of SWI/SNF related, matrix associated, actin dependent regulator of chromatin, subfamily b, member 1 (Smarcb1) was found to be down-regulated. Conversely, an increase in the expression levels of SLIT and NTRK-like family, member 1 (Slitrk1), TYRO3 protein tyrosine kinase 3 (Tyro3), epilepsy, progressive myoclonic epilepsy, type 2 gene alpha (Epm2a), fucosyl-transferase 9 (Fut9), immunoglobulin superfamily, member 9B (Igsf9b) was observed. These genes are known to play crucial roles in nervous system development (Table 1).

#### 2.3.2. Positive Regulation of Synaptic Transmission

PSTs primarily exert their effects on the presynaptic membrane by binding to toxin receptors located on its surface. Our findings demonstrate a down-regulation of proteins, such as MARCKS-like 1 (Marcksl1) and glutamate receptor, ionotropic, kainate 2 (beta 2) (Grik2), which are localized in the presynaptic membrane. Conversely, ephrin B3 (Efnb3) and fragile X messenger ribonucleoprotein 1 (Fmr1) exhibited an increase in expression levels. Additionally, discs large MAGUK scaffold protein 5 (Dlg5) and dystrobrevin, beta (Dtnb), located in the postsynaptic membrane, increased (Table 1).

#### 2.3.3. Ion Transport

Alterations in ion transport proteins were observed in the model group. The findings revealed that MRS2 magnesium transporter (Mrs2), leucine rich repeat containing 8 family, member C (Lrrc8c), mucolipin 1 (Mcoln1), solute carrier family 23 (nucleobase transporters), member 2 (Slc23a2), solute carrier family 38, member 3 (Slc38a3), store-operated calcium entry-associated regulatory factor (Saraf) exhibited up-regulation. Additionally, potassium inwardly rectifying channel, subfamily J, member 11 (Kcnj11), potassium inwardly-rectifying channel, subfamily J, member 9 (Kcnj9), potassium voltage-gated channel, shaker-related subfamily, beta member 3 (Kcnab3), sodium channel, voltage-gated, type I, alpha (Scn1a) demonstrated significant up-regulation, with close association also to voltage-gated ion channel activity (Table 1).

#### 2.3.4. Transferase Activity

PSTs exert an impact on transferase activity in the brain of mice. The levels of several proteins, including MGAT4 family, member C (Mgat4c), N(alpha)-acetyltransferase 30, NatC catalytic subunit (Naa30), deltex 3, E3 ubiquitin ligase (Dtx3), fucosyl-transferase 9 (Fut9) were found to be increased in the model group. Conversely, EEF1A alpha lysine methyltransferase 1 (Eef1akmt1), glutathione S-transferase, theta 1 (Gstt1) and protein phosphatase 1, regulatory inhibitor subunit 11 (Ppp1r11) exhibited down-regulation. Notably, alterations in kinase expression suggested a potential role in phosphorylation regulation. Specifically, N-acetylglucosamine kinase (Nagk), TYRO3 protein tyrosine kinase 3 (Tyro3), cytidine monophosphate (UMP-CMP) kinase 2, mitochondrial (Cmpk2), sphingosine kinase 2 (Sphk2) and unc-51 like kinase 2 (Ulk2), were up-regulated in the model mice. On the other hand, seleno-phosphate synthetase 2 (Sephs2) was observed to be down-regulated (Table 1).

## 3. Discussion

The guanidine neurotoxin STX, a sodium channel blocker, is one of the main toxins in PSTs. The World Health Organization (WHO) has established a limit of 80 μg or 400 MU/100 g for PSTs in an edible portion of shellfish weighing 100 g. Currently, most countries regulate and monitor shellfish based on this standard. PST poisoning carries a high mortality rate, with toxic doses ranging from 600–5000 MU and lethal doses ranging from 3000–30,000 MU. However, the comprehensive proteomic alterations in the entire brain of model mice induced by STX remain unclear. In this study, adult male mice were administered STX to establish a chronic low-dose shellfish poisoning model, which was subsequently confirmed using behavioral tests and histological findings. To investigate large-scale variations in the entire brain proteome induced by chronic low-level shellfish poisoning caused by PSTs, we conducted a proteomics study using TMT combined with LC–MS. The altered proteins primarily participated in the development of the nervous system, positive regulation of synaptic transmission, ion transport and transferase activity. These processes may be implicated in the pathogenesis of chronic low-dose shellfish poisoning. These findings are significant and warrant further investigation. PSTs are considered classical neurotoxic agents. Impairment of cognitive ability in mice during the Morris water maze test demonstrated damage to the nervous system in the shellfish poisoning model group. Previous studies have indicated that Tyro3 signaling plays a crucial role in various processes, such as protecting neurons from excitotoxic injury, promoting platelet aggregation, and facilitating skeleton reorganization [14]. Fut9 is involved in Lewis × (Lex)/CD15 epitope biosynthesis in neurons, which enables cell differentiation, cell adhesion, and initiation of neurite outgrowth [15]. Similarly, up-regulation of Tyro3 and Fut9 was observed in the model group, indicating perturbation to the nervous system upon stimulation by PSTs.

Evidence suggests that PSTs inhibit nerve conduction by modulating sodium ion channels, which are abundant in synapses. Proteomic analysis revealed alterations in synaptic transmission, including the up-regulation of Efnb3, Fmr1, Dlg5 and Dtnb, as well as the down-regulation of Marcksl1 and Grik2. The phosphorylation of Marcksl1 by MAPK8 induces the formation and stabilization of actin bundles, thereby reducing actin plasticity and restricting neuronal migration [16]. L-glutamate acts as an excitatory neurotransmitter at many synapses in the central nervous system. Grik2 functions as an ionotropic glutamate receptor that binds to L-glutamate, inducing a conformational change that opens the cation channel and converts chemical signals into electrical impulses [17]. Efnb3 may play a crucial role in forebrain function by binding to commissural axons/growth cones and inducing their collapse. Fmr1 negatively regulates the voltage-dependent calcium channel current density in soma and presynaptic terminals of dorsal root ganglion (DRG) neurons, thus regulating synaptic vesicle exocytosis [18]. Dlg5 plays a crucial role in the formation of dendritic spine and synaptogenesis in cortical neurons, as it facilitates the cell surface localization of N-cadherin to regulate synaptogenesis. Additionally, Dtnb may be implicated in the regulation of cell proliferation during the early stages of neural differentiation [19]. Based on these proteomic findings, we hypothesize that low-dose PST stimulation inhibits sodium ion channels, leading to compensatory mechanisms within the mice nervous system that enhance neuronal proliferation and migration, while also regulating synaptogenesis. However, further experiments are necessary to validate this hypothesis.

The occurrence of shellfish poisoning induced by PSTs also led to the modification of transferase activity proteins. Naa30 catalyzes the acetylation of peptides with N-terminal methionine residues, specifically those starting with Met-Leu-Ala and Met-Leu-Gly. Dtx, functioning as a ubiquitin ligase protein, regulates the Notch pathway through its ubiquitin ligase activity [20]. Phosphorylation is a prevalent and significant posttranslational modification that governs various biological activities, such as cell apoptosis and cell cycle regulation. Studies have shown that okadaic acid, the primary cause of diarrheal shellfish poisoning, is associated with neuronal apoptosis, tau protein hyperphosphorylation, and morphological changes by modulating protein phosphorylation [21]. The shellfish poisoning model also exhibited an evident increase in protein phosphorylation. Cmpk2 is involved in the synthesis of dUTP and dCTP with mitochondria, while displaying a wide range of nucleoside diphosphate kinase activity. Sphk2 catalyzes the phosphorylation of sphingosine to generate sphingosine-1-phosphate, which plays a crucial role in promoting mitochondrial functions by regulating ATP and ROS levels in dopaminergic neurons [22]. Ulk2 Acts upstream of phosphatidylinositol 3-kinase PIK3C3 to regulate auto-phagophore formation; it is activated early during neuronal differentiation through AMPK-mediated phosphorylation. Nagk converts endogenous N-acetylglucosamine (GlcNAc), a major component of complex carbohydrates derived from lysosomal degradation or nutritional sources, into GlcNAc 6-phosphate [23].

In summary, a successful induction of a shellfish poisoning model was achieved in Kunming mice through oral administration of PSTs. Prolonged exposure to low-dose PSTs for 10 weeks resulted in significant cognitive deficits. LC–MS analysis using TMT revealed 123 altered proteins in the model group compared to the normal group. GO annotation indicated significant regulation of clusters related to nervous system development, positive regulation of synaptic transmission, and significant regulation of ion transport and transferase activity. Overall, this study investigated proteomic changes in a shellfish poisoning-induced nerve damage model, and these findings provide new insights and evidence for paralytic shellfish toxins neurotoxicity. Further investigations are required to elucidate the role of identified key proteins in shellfish poisoning-mediated cognitive impairment and the associated molecular mechanisms.

## 4. Materials and Methods

### 4.1. Animals

Kunming mice (Specific Pathogen Free degree, aged 5 weeks and weighing 18 ± 2 g) were procured from Beijing Jinmuyang Experimental Animal Breeding Co. Ltd. (Beijing, China). The mice were maintained in a temperature-controlled condition of 22 ± 2 °C, 40–60% relative humidity and a 12 h light/dark cycle. The mice had ad libitum access to food and water. Mice were randomly divided into two groups, a control group (*n* = 15) and a model group (*n* = 15). The control group received a standard diet and drinking water, while the model group also received a standard diet and drinking water (5 μg STX/kg body/day). Water intake was measured weekly in the model group to prepare fresh aqueous STX solution accordingly. STX was obtained from the National Research Council (NRC) of the National Measurement Standards Institute of Canada. Behavioral tests assessing cognitive function were conducted on both control mice and model mice during the tenth week of administration. At the end of this period, all mice were euthanized by cervical dislocation method for sample collection, including serum and brain samples for further analysis purposes. All animal experiments in this study were approved by the ethics committee of State Key Laboratory of NBC Protection for Civilians (NO. LAE-202-001).

### 4.2. Behavioral Tests

Considering the experimental interaction, the mice were sequentially subjected to step-down passive avoidance test (SDPA) and Morris water maze test (MWM), respectively. A 2-day interval was maintained between each behavioral test. The behavioral tests were conducted in an air-regulated and soundproof experimental laboratory between 9:00 and 17:00. After testing, the apparatus was cleaned with 70% ethanol and water to remove olfactory traces.

PNT: over the course of the subsequent 5 days, the mice were trained to find the hidden platform for 4 × 90-s trials per day. After gently guiding them to the platform, they stayed there for 10 s and then returned to the cage. If they could not find the platform themselves, they would be manually placed on it. The Morris water maze test was carried out as described elsewhere [24].

SPT: during the probe test, mice were subjected to a 120 s swimming session in a pool without an escape platform. Various parameters included escape latency, number of crossings, time spent in the target quadrant, and distance traveled in the target quadrant, The Morris water maze test was carried out as described elsewhere [25].

### 4.3. Proteomic Analysis

Sample preparation: Four brain tissue samples were selected from each group, with the control group’s samples labeled as control-1, control-2, control-3 and control-4, and the model group’s labeled as model-1, model-2, model-3 and model-4. The entire brain tissue was resuspended and homogenated in RIPA lysis buffer (ID: P0013B, Beyotime, Shanghai, China). After 2 min of 30 Hz ultrasound in the ice bath, the lysate was centrifuged at 12,000× *g* for 10 min at 4 °C and the supernatant was retained (Eppendorf, Hamburg, Germany). The protein concentration of each sample was determined using a BCA protein assay kit (ID: A045-4, Jiancheng, Nanjing, China), according to the instructions of the manufacturer, and the samples were stored in a −20 °C refrigerator for use.

#### 4.3.1. Protein Digestion and TMT Labeling

A total of 100 μg proteins from each group were diluted with H_2_O to 1 mg/mL. Acetone was pre-cooled to −20 °C, 5 times the volume of acetone was added to the protein samples, mixed and precipitated at −20 °C overnight (ANPEL Laboratory Technologies, Shanghai, China). The suspension was centrifuged at 12,000× *g* for 10 min at 4 °C and the pellet collected (Eppendorf, Hamburg, Germany). 100 μL protein resolve buffer was added and the protein pellet was dissolved by ultrasound for 3 min, with the addition of 5 mM DTT (final concentration) incubating at 55 °C for 20 min and 15 mM IAA (final concentration) for 1 h in the dark. After reduction and alkylation, samples were digested by trypsin (Promega, WI, USA) at 37 °C for 16 h, with ratios of protein to trypsin of 50:1, and labeled using the TMT Multiplex Kit (10-plex, Thermo, Waltham, MA, USA) according to the manufacturer’s protocol. Four biological replicates were performed for the control group, and four biological replicates were used for the model group. The control group samples were labeled as 127N, 128N, 129N and 130N, while model group samples were labeled as 127C, 128C, 129C and 130C, respectively.

#### 4.3.2. High pH Reversed Phase (RP) Chromatography Fractionation

TMT-labeled peptides were fractionated using the previously described method [17]. Before high performance liquid chromatography (HPLC) analysis, equal amounts of 100 μL of the TMT-labeled peptide of different moieties were mixed and desalted with TFA. Add TFA to the mixed sample (final concentration 2%) and centrifuge at 13,000 rpm for 10 min and collect the supernatant. In the binary solvent system, the mobile phase A contained 10 mM Ammonium acetate in water (pH 10.0), whereas mobile phase B contained 10 mM ammonium acetate and 90% ACN (*v*/*v*, pH 10.0). Peptides were loaded directly onto the Xbridge BEH C18 XP Column for the C18 analytical runs (Waters, Milford, MA, USA) followed by HPLC separation (UltiMate 3000 UHPLC; Thermo Fisher Scientific, Waltham, MA, USA). LC separation was performed at a flow rate of 500 μL/min using a linear 60 min gradient of 5% B over 2 min, 5–30% B over 40 min, 30–40% B over 10 min, 40–90% B over 4 min, 90% B over 2 min and 90–5% B over 2 min. Fractions were collected and then numbered in sequence. In the first 12 min, one minute was divided into one component, named 1–12 component. The effluent at 13 min was inserted into 1 component, the effluent at 14 min was inserted into 2 component, the effluent at 15 min was inserted into 3 component, etc., collected in cycles, and combined into 12 fractions, which were vacuum-dried for subsequent experiments. This is carried out so that more proteins can be identified.

#### 4.3.3. Nano LC–MS/MS Analysis

The fractionated peptides were analyzed using a Q Exactive HFX (Thermo Fisher, MA, USA) coupled with nanoUPLC EASY-nLC1200 (Thermo Fisher, MA, USA). Mobile phase A contained 0.1% formic acid and 2% ACN (*v*/*v*), while mobile phase B contained 0.1% formic acid and 80% ACN (*v*/*v*). Peptides were separated using ReprosilPur 120 C18 AQ column (Dr. Maisch, Tübingen, Germany) at an eluent flow rate of 300 nL/min using a linear gradient over 90 min. The gradient comprised the following steps: 2–5% B over 2 min, 5–22% B over 68 min, 22–45% B over 16 min, 45–95% B over 2 min, and 95% B over 2 min. Data dependent acquisition (DDA) was performed in profile and positive mode with Orbitrap analyzer at a resolution of 120,000 (@200 m/z) and *m*/*z* range of 350–1600 for MS1; for MS2, the resolution was set to 45 k with a fixed first mass of 110 *m*/*z*. The automatic gain control (AGC) target for MS1 was set to 3E6 with max IT 30 ms, and 1E5 for MS2 with max IT 96 ms. The top 20 most intense ions were fragmented by HCD with normalized collision energy (NCE) of 32%, and isolation window of 0.7 *m*/*z*. The dynamic exclusion time window was 45 s, and single charged peaks and peaks with charge exceeding 6 were excluded from the DDA procedure.

#### 4.3.4. Data Analysis

Vendor’s raw MS files were processed using Proteome Discoverer(PD) software (Version 2.4.0.305) and the built-in Sequest HT search engine. MS spectra lists were searched against their species level UniProt FASTA databases (uniprot-Mus+musculus-10090-2021-8.fasta, 17070 reviewed entries, downloaded 1 August 2021), with Carbamidomethyl (C), TMT 6 plex (K) and TMT 6 plex (N-term) as a fixed modification and Oxidation (M) and Acetyl (Protein N term) as variable modifications. Trypsin was used as protease. A maximum of 2 missed cleavage(s) was allowed. The false discovery rate (FDR) was set to 0.01 for both PSM and peptide levels. Peptide identification was performed with an initial precursor mass deviation of up to 10 ppm and a fragment mass deviation of 0.02 Da. Unique peptide and Razor peptide were used for protein quantification and total peptide amount for normalization. All the other parameters were reserved as default. For protein-abundance ratios measured using TMT, differentially expressed proteins were determined (requirements: *p* < 0.05, fold change <0.83 or >1.2). Bioinformatic analysis was performed using the OmicShare tools at https://www.omicshare.com/tools, accessed on 1 April 2023. All proteins identified in the mouse hippocampus were characterized using DAVID Bioinformatics Resources 6.8 (https://david.ncifcrf.gov/, accessed on 13 April 2023) and Omic Share tool (https://www.omicshare.com/tools, accessed on 3 April 2023) for molecular function (MF), cellular component (CC), biological process (BP), and Kyoto Encyclopedia of Genes and Genomes (KEGG) pathway classification.

### 4.4. Statistical Analysis

The significance values were calculated using SPSS 22.0 software. Data were analyzed by homogeneity of variances and one-way analysis of variance. *p*-values less than 0.05 were considered statistically significant.

## Figures and Tables

**Figure 1 marinedrugs-22-00108-f001:**
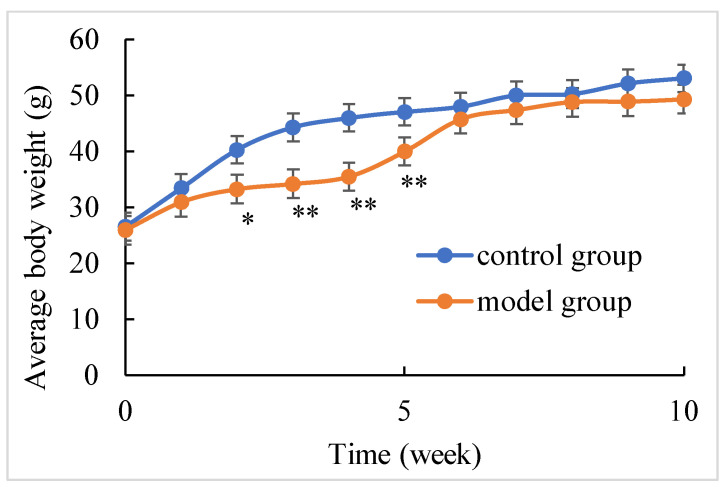
Body weight of control and chronic shellfish poisoning model mice at ten weeks. (* *p* < 0.05, ** *p* < 0.01).

**Figure 2 marinedrugs-22-00108-f002:**
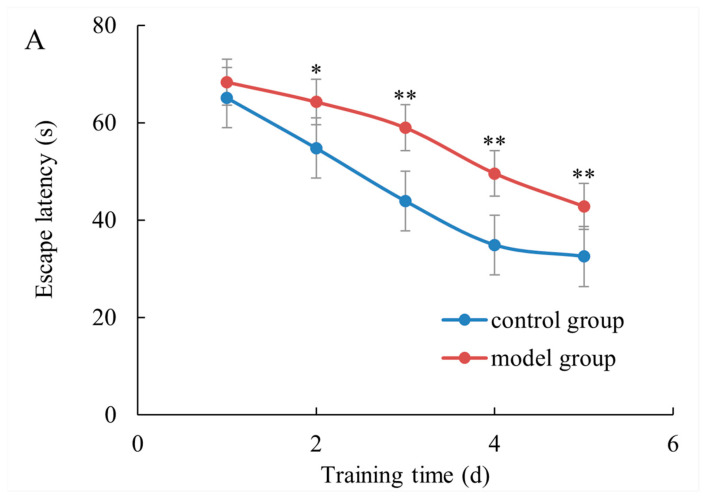
Shellfish poisoning impaired cognitive performance of mice. (**A**) The escape latency of mice in a training session from day 1 to day 5. (**B**–**D**) Differences in probe time, the number of crossing movements, target quadrant time percentage, target quadrant distance percentage and platform crossing times in the probe trial of the MWM test. * *p* < 0.05, ** *p* < 0.01.

**Figure 3 marinedrugs-22-00108-f003:**
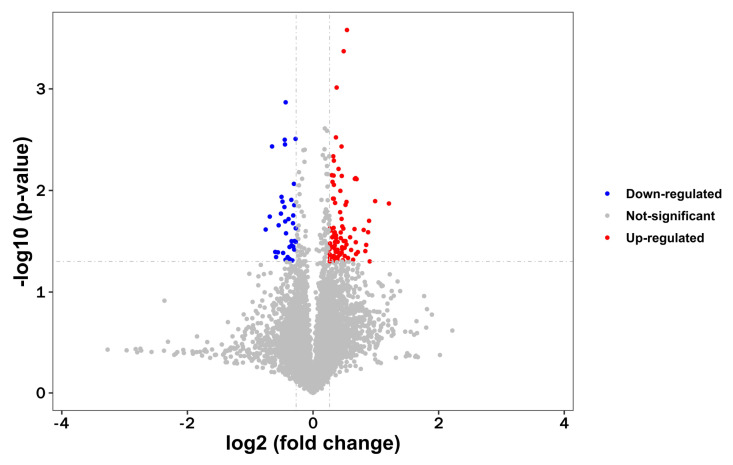
Volcano plot of differentially expressed proteins between control and shellfish poisoning model groups. Volcano plot of ratios and *p*-value representing protein abundance changes between control and model groups. Proteins with *p* < 0.05 and above/below 1.2-fold changes are identified as proteins with significant changes. Down-regulated, *p* < 0.05 (blue dots); up-regulated, *p* < 0.05 (red dots).

**Figure 4 marinedrugs-22-00108-f004:**
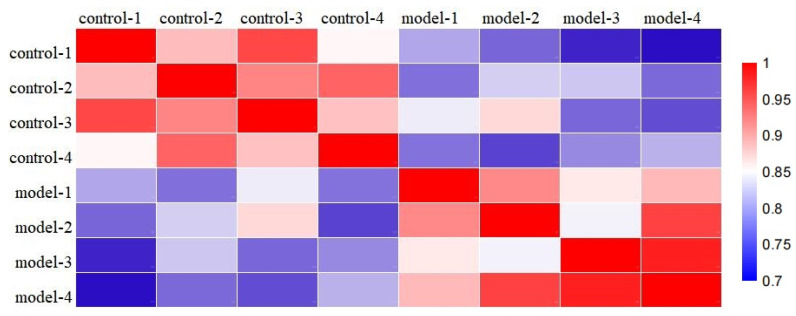
Correlation of the protein intensity among four biological replicates of control and shellfish poisoning model groups, as analyzed with R language. The Pearson color scale bar is shown (right).

**Figure 5 marinedrugs-22-00108-f005:**
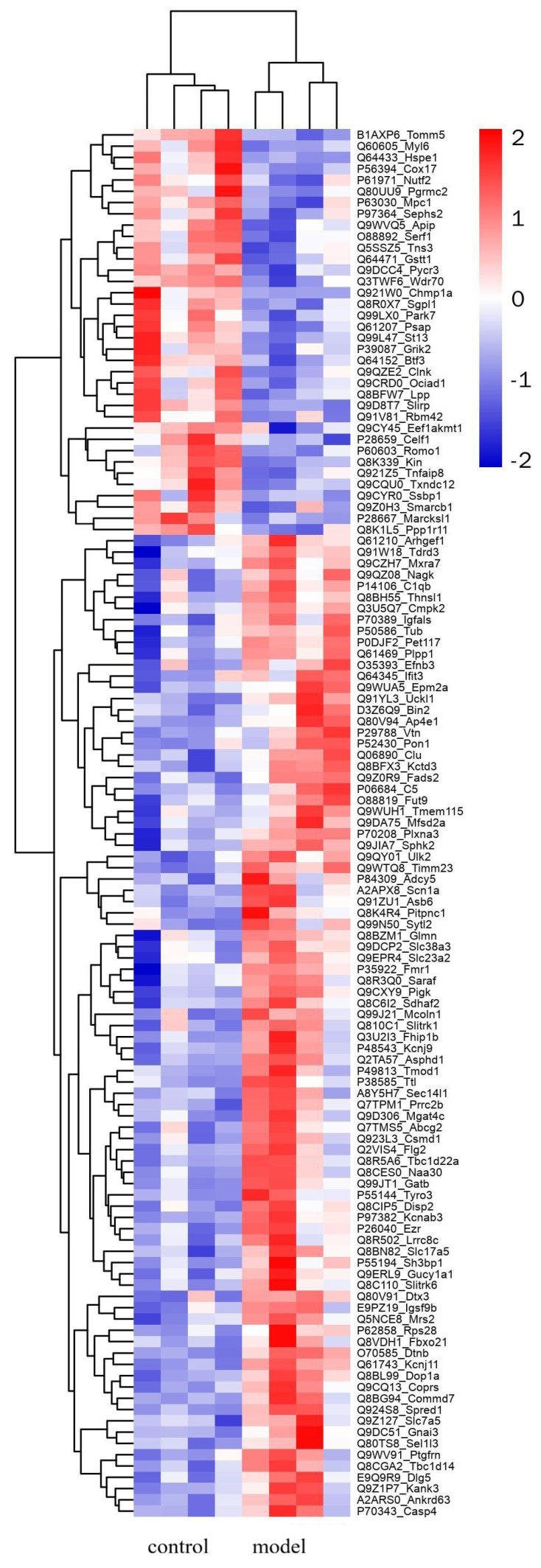
Hierarchical cluster analysis of differentially expressed proteins between control and shellfish poisoning model groups. Rows represent each protein, and columns different groups. Color indicates relative expression of protein: red represents higher expression, and blue represents lower expression.

**Figure 6 marinedrugs-22-00108-f006:**
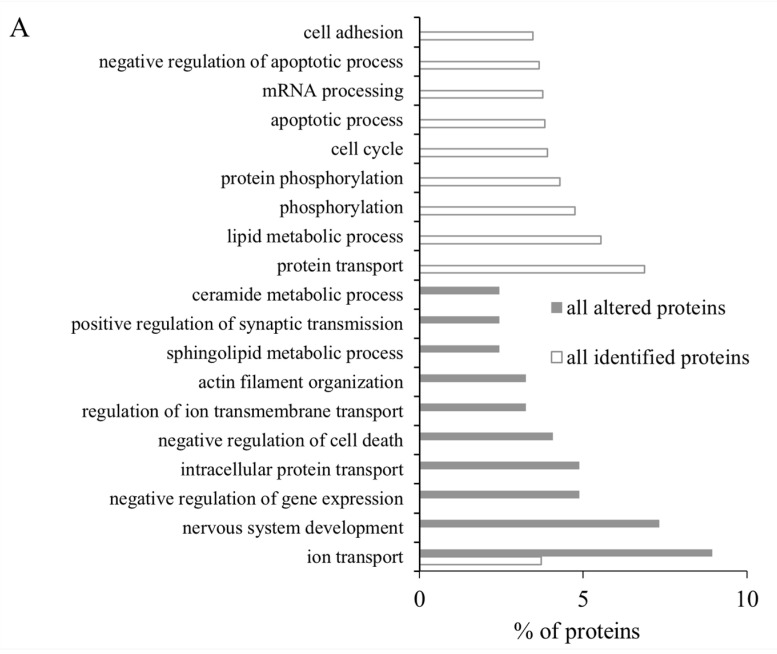
GO classification of all identified proteins and all altered proteins between control and shellfish poisoning model groups. (**A**) Biological process of all identified proteins and all altered proteins in brain; (**B**) molecular function of all identified proteins and all altered proteins in brain; (**C**) cellular component assigned to all identified proteins and all altered proteins in brain. The first 10 subtypes of all identified proteins and all altered proteins in GO classification were analyzed.

**Figure 7 marinedrugs-22-00108-f007:**
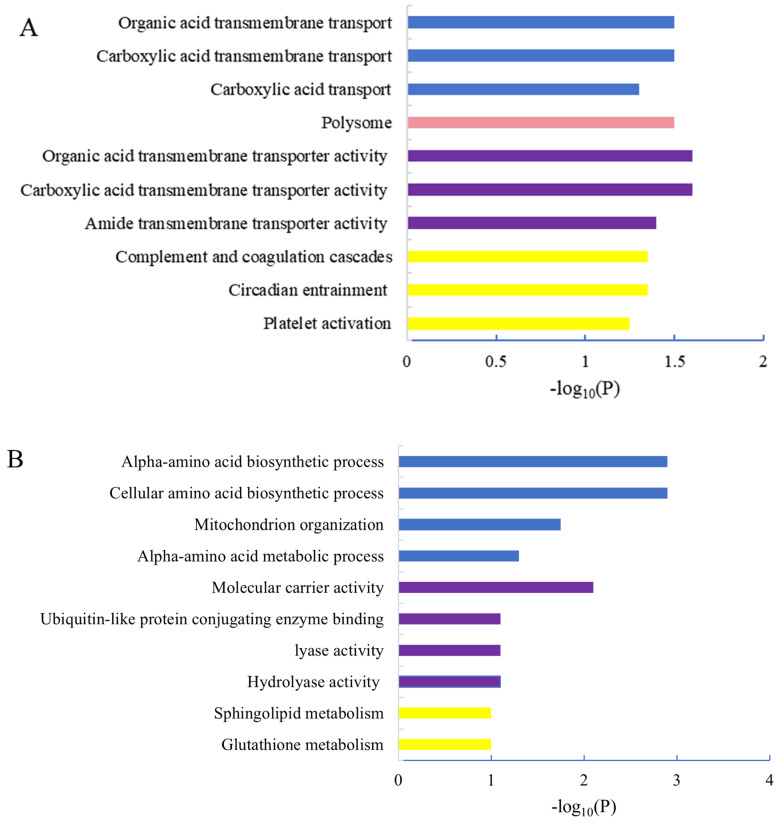
GO classification of up-regulated altered proteins and down-regulated altered proteins between control and shellfish poisoning model groups. (**A**) GO classification of up-regulated proteins; (**B**) GO classification of down-regulated proteins. Blue is BP, pink is CC, purple is MF and yellow is KEGG.

**Table 1 marinedrugs-22-00108-t001:** List of representative proteins where expression content significantly changed in control and shellfish poisoning model groups.

Name	Accession	GeneName	*p* Value	Fold Change
Sodium channel protein type 1 subunit alpha	A2APX8	Scn1a	0.05	1.27
UMP-CMP kinase 2, mitochondrial	Q3U5Q7	Cmpk2	0.04	1.35
Glutamate receptor ionotropic, kainate 2	P39087	Grik2	0.05	0.80
Sodium-coupled neutral amino acid transporter 3	Q9DCP2	Slc38a3	0.03	1.36
N-acetyl-D-glucosamine kinase	Q9QZ08	Nagk	0.04	1.27
Synaptic functional regulator FMR1	P35922	Fmr1	0.03	1.36
Solute carrier family 23 member 2	Q9EPR4	Slc23a2	0.04	1.38
MARCKS-related protein	P28667	Marcksl1	0.01	0.71
Glutathione S-transferase theta-1	Q64471	Gstt1	0.02	0.59
Dystrobrevin beta	O70585	Dtnb	0.00	1.40
Protein turtle homolog B	E9PZ19	Igsf9b	0.04	1.20
Volume-regulated anion channel subunit LRRC8C	Q8R502	Lrrc8c	0.05	1.31
Sphingosine kinase 2	Q9JIA7	Sphk2	0.01	1.32
E3 ubiquitin-protein ligase PPP1R11	Q8K1L5	Ppp1r11	0.02	0.62
EEF1A lysine methyltransferase 1	Q9CY45	Eef1akmt1	0.05	0.77
Magnesium transporter MRS2 homolog, mitochondrial	Q5NCE8	Mrs2	0.01	1.45
Probable E3 ubiquitin-protein ligase DTX3	Q80V91	Dtx3	0.01	1.28
N-alpha-acetyltransferase 30	Q8CES0	Naa30	0.03	1.31
SWI/SNF-related matrix-associated actin-dependent regulator of chromatin subfamily B member 1	Q9Z0H3	Smarcb1	0.04	0.81
G protein-activated inward rectifier potassium channel 3	P48543	Kcnj9	0.02	1.40
SLIT and NTRK-like protein 1	Q810C1	Slitrk1	0.05	1.47
Tyrosine-protein kinase receptor TYRO3	P55144	Tyro3	0.04	1.64
Serine/threonine-protein kinase ULK2	Q9QY01	Ulk2	0.01	1.37
ATP-sensitive inward rectifier potassium channel 11	Q61743	Kcnj11	0.00	1.45
Store-operated calcium entry-associated regulatory factor	Q8R3Q0	Saraf	0.04	1.35
Selenide, water dikinase 2	P97364	Sephs2	0.04	0.68
Ephrin-B3	O35393	Efnb3	0.05	1.39
Disks large homolog 5	E9Q9R9	Dlg5	0.02	1.21
Laforin	Q9WUA5	Epm2a	0.03	1.25
Mucolipin-1	Q99J21	Mcoln1	0.03	1.21
Voltage-gated potassium channel subunit beta-3	P97382	Kcnab3	0.02	1.58
4-galactosyl-N-acetylglucosaminide 3-alpha-L-fucosyltransferase 9	O88819	Fut9	0.02	1.35
Alpha-1,3-mannosyl-glycoprotein 4-beta-N-acetylglucosaminyltransferase C	Q9D306	Mgat4c	0.04	1.61

## Data Availability

The original data presented in the study are included in the article/Appendix A; further inquiries can be directed to the corresponding author.

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
