# Peer review of "Differential Proteomic Analysis of Low-Dose Chronic Paralytic Shellfish Poisoning"

_marinedrugs, 2024, doi:10.3390/md22030108_

Round 1

Reviewer 1 Report

Comments and Suggestions for Authors

The topic of the manuscript is of undoubted interest:  on a mouse model  the authors compared  the behavioral effects  of the shellfish poisoning  with the changes in the  proteome in  various parts of the brain and  on this basis made interesting conclusions concerning  which steps in the signaling cascades might be affected  and would result  in the disturbed neuronal activities.

However, I have several critical remarks.

1.The authors   are using term “ altered proteins”  for those proteins for which some differences in their amounts were found by proteomics  - the term is clearly not good since might mean proteins with the  changes in the amino acid sequence, post-translational modifications or any other properties.

2. Many abbreviations are introduced,  often absolutely unnecessary (like CC-cellular compound) and there is no list of those abbreviations in the manuscript.

Fig.6 it is not clear where are control and where  the  experimental group.

Table 1. Should be a list of proteins  for which the measured amounts significantly changed.

3.3.2. Please, stress that the amount was changed

2.3.3. “enhanced ion transport was  observed” -it is not correct: increase in the amount  of transporters and other proteins was observed, but there was no measurements of the enhanced transport as such.

2.3.4. the same concerns the ion transport

Line 220-223 The sentence “ The findings are meaningful and require further research” is repeated twice. “ The  PST are considered classical nerve agents”  is wrong with the chosen abbreviations for PST.

226- give a reference to the mentioned increased phosphorylation

In Fig. 6C for Golgi apparatus are indicated all identified proteins (meaning that no altered proteins  were  detected , which can be understood), while  for presynaptic membrane  are shown only all altered proteins (presuming the absence of all identified proteins?)

Strange sentences:

Line 357 100 microL protein resolution was added?

377 Fraction collected and then was measured in sequence?

“one component was collected for every minute in one tube”  -and what if two or more  components would  elute during one minute???

357 “ 100 microL protein resolution was added” – what does it mean?

Comments on the Quality of English Language

In general, English  is accetable.

Reviewer 2 Report

Comments and Suggestions for Authors

I have carefully reviewed the manuscript entitled "Differential proteomic analysis of low-dose chronic shellfish poisoning". The study aimed to investigate proteomic changes in mouse brains related to chronic saxitoxin (STX) poisoning, which provides valuable insights. However, there are some methodological and reporting issues that require addressing before this work can be considered for publication. I have the following comments and suggestions for improving the manuscript:

Introduction:

Reference 5 cited in the text is not present in the reference list. Please verify and add the missing reference.

The term "STX" first appears on line 46 but is defined later on line 59. For clarity and flow, please define abbreviations at the first instance.

Most of the background information cited focuses on acute shellfish poisoning (e.g. reference 7). Consider providing more context on the significance of studying chronic shellfish poisoning effects to adequately rationale the study.

Results and Discussion:

The experiment investigated the effects of STX (the major group of PST), however the term "PST" is used in the Results and Discussion Section which lacks accuracy.

It would strengthen the analysis to report the results of principal component analysis or relevant tests to demonstrate the quantitative proteomics data sufficiently explains variance between treatment and control groups.

The rationale for gene ontology annotation could be clarified. Consider focusing on gene ontology enrichment analysis rather than raw count data, as enrichment better compares actual to expected occurrences for meaningful biological interpretation.

The color coding is not consistent between Figures 7A and 7B, for example molecular function is labeled purple in 7A but red in 7B. Please reconcile for consistency.

The title of Table 1 "List of proteins were significantly changed in control and shellfish poisoning model groups" is misleading as it does not accurately represent the contents. While the analysis identified 123 proteins with significant abundance changes, the table only lists 32 proteins. Please revise the table title to clarify it does not include all significantly changed proteins.

A heatmap, bar chart or other graphical representation would be more informative than a table for visualizing expression patterns of differentially expressed proteins.

While protein abundance changes for transferase were observed, experiments assessing transferase enzymatic activity were not reported. Please qualify conclusions about potential activity changes accordingly instead of stating outright reductions.

Material and Methods:

Line 348 – 350: The statement "the suspension was centrifuged at 12000 × g for 10 min at 4°C and collected" is ambiguous. Please clarify whether you mean "The lysate was centrifuged at 12000 × g for 10 min at 4°C and the supernatant was retained".

Line 352: The statement "The samples were collected for use" is ambiguous. Please specify the storage conditions of the samples before use.

Line 354: The meaning of the statement "A total of 100 μg proteins from each group was reduced with acetone" is ambiguous. Please clarify what reduction with acetone entails in this context.

Line 357: Please define the term "protein resolution" as it is used here.

Line 358: The conditions (duration and temperature) for DTT treatment are missing. Please specify the treatment conditions.

Line 359: After reduction and alkylation, the sample should be in liquid form. Please clarify what is meant by the statement "then the pellet was collected again".

Line 362: Please specify which TMT kit was used (e.g. 16-plex, 18-plex, etc.).

Line 364 – 365: The number labels for TMT labeling should indicate the isobaric mass with C or N, not just the number alone. Please revise accordingly.

Line 368 – 369: The statement "equal amounts 100μL of the TMT-labeled peptide of different moieties were mixed and desalted with TFA" requires clarification. What is the meaning of “desalted with TFA”.

Line 378 – 379: Please specify your criteria for combining mass spectrometry fractions.

Line 389: Please clarify if you mean an m/z range of 35-1600.

Line 399: Please specify when the proteome database was acquired and whether it contains isoform sequences and only reviewed sequences.

Line 403: Please specify whether target decoy or fixed value decoy was used for FDR control and the method used to calculate FDR.

Line 408: Please specify the type of adjustment used for adjusted P-values.

Line 417 - 418: Please specify which statistical test(s) was used to calculate significance values in SPSS 22.0 software.

Comments on the Quality of English Language

Here are my suggestions to improve the English usage and clarity:

1. Sentence structure follows patterns that are more common in other languages than English in some places. For example:

"Morphological examination of H&E-stained brain tissues in different groups were examined to evaluate his pathologic variaiton." This sentence would be structured as "H&E-stained brain tissues in different groups were examined to evaluate their pathological variation." Using "his" rather than "their" and putting the verb before the subject is more common in some other languages.

"During the experiment, no death occurred in the control group and the model group." In this case, the typical English structure would be "During the experiment, no deaths occurred in the control group or the model group." Placing both clauses directly next to each other without connecting words like "or" is characteristic of some other languages.

"After mixing, overnight precipitation at -20°C." the typical English structure would be "After mixing, precipitation overnight at -20°C.". etc.

2.  Some phrasing is unclear and hindrance understanding. For example:

"Collection these fractions in cycles, combined into 12 fractions", the phasing “Collection these fractions in cycles” is unclear.

Reviewer 3 Report

Comments and Suggestions for Authors

Dear Editor, Greetings.

I report that I have read in detail the article entitled "Differential proteomic analysis of low-dose chronic shellfish poisoning" by Liu et al.

The purpose of this study was to address whether chronic asymptomatic shellfish poisoning leads to potential neurotoxicity and health effects in the mammalian system.

However, the article is nuanced and also unclear as to the causes and effects produced by the saxitoxin-group. The end point of the proposal is clearly understood, but the toxin basis of the study is not very clear and orienting. The authors should clarify concepts to argue well the toxicology proposal, as well as the experimental procedures included.

I suggest giving the paper an extensive revision for further review.

Comments:

Line 8: What is TMT?

Line 23: Use the term Harmful algae blooms (HAB) and not red tide.

Introduction

I suggest to give an orientation of this section more related to the topic. The first part of this section dilutes in interest since it covers other aspects that are not related to the subject proposed by the authors. It would be more interesting the proposal in relation to the described analogues, the toxic variability, the capacity of modification of the profiles. Concepts that complement and help to understand intoxication.

It is also necessary to be more precise in the level of action of the toxins on the voltage-dependent sodium channels and the interaction between the type of channel and the toxins.

Another important aspect is to define in detail the toxic effects and causes of death, using the appropriate references. It is important to highlight in previous studies the routes of administration and doses used to make a real comparison.

It is important to discuss your results based on the other saxitoxin group analogs that can produce toxic pictures and the relationship between acute reference doses (ARfD) and toxic equivalency factors (TEF).

Line 61-65: Please review this sentence in detail.

Line 67-77: Be more precise with the objective and hypothesis stated.

Fig 1 & 2: Be more precise in the description of the figures.

Line 81-95: Why exclude an analysis of other organs?. According to the effects you have included, some differences should occur at the respiratory (lung) level rather than the brain. What is the effective dose to cross the blood-brain barrier?

Line 101: Clarify what is PNT?

Line 101-122: The toxic effects could be more related to a muscular blockade, rather than a cognitive one. Please clarify this point!

Line 120: What toxicity and/or concentration allows to define neurotoxicity?

Line 208-210: Please review this sentence in detail.

Line 211: Include the references.

Line 211-213: Rewrite the sentence. Use correct units and include references.

Line 234-235: Rewrite the sentence and include the references.

Line 250-251: Please clarify what type of channels the STX analogs act on.

Line 263_265: You are using as a basis toxins totally opposite to the chemistry and level of action of STX. Delete this section.

Line 277: What is low dose?

Line 296: What data allow you to establish the use of this dose of toxins?

Line 297-299: How do you ensure the stability of the toxins in the water?

What method do you use to analyze the toxins?

Line 337-342: Please be more precise in your methodology.

Comments on the Quality of English Language

Moderate editing of English language required.

Reviewer 4 Report

Comments and Suggestions for Authors

This manuscript is devoted to the study of low-dose poisoning with PSP toxins. The authors, using proteomic analysis, showed that this group of toxins affects the nervous system of animals. For TMT-based differential proteomic analysis, I have no comments on them. My main remark concerns the histological examination.

Major concern:

Line 87-92. Authors told about degenerative processes in brain tissue and referred to the Figure 2. I would like to point that the both images are looked similar. The resolution of the Figure 2 does not give me to verify their proposition (I mean “granulovacuolar degeneration”). Please set the images with more higher resolutions and point to the “granulovacuolar degeneration” by the arrows.

Minor concern:

Figure 2. Please, set scale bur instead magnification.

Round 2

Reviewer 1 Report

Comments and Suggestions for Authors

I am glad that the authors  considered all my questions and  critical comments and introduced the respective changes.

My first question concerned the term “ altered proteins “ and the authors answered that this term is accepted in proteomics  and  is related to the proteins with the increased or decreased content. However, the manuscript is addressed not only to the “proteomics audience”:  the authors are also dealing with the effects  of the toxins on the behavior and they should clearly explain this term mentioning the alternating levels, upregulated and downregulated proteins. They should also mention whether they detected any post-translational modifications.

The whole text needs attentive English editing, among most often mistakes  are inconsistencies  in verbs for singular and plural cases.

I still consider that there are   two many abbreviations -at least  the list of them should be available.

Line 74 -where is the reference for Sun et al?

Fig.6. Still do not understand how in some cases (for example, membrane) the number of altered proteins can be larger than that of the identified proteins

Line 191 should be “where” the expression content….

Line 381.What does it mean “ was desalted  with TFA” ? Why then it is written as an instruction “ Add TFA etc?”

Line 390 should be “ fractions were collected….”

Comments on the Quality of English Language

I have already mentioned the need of English editing in  the text to the authors.

Author Response

We are thankful to the reviewer for valuable suggestions to that aided in improving our manuscript. Here, we respond to the comments in the queries and suggestions.

------

I am glad that the authors considered all my questions and critical comments and introduced the respective changes.

  1. My first question concerned the term “ altered proteins “ and the authors answered that this term is accepted in proteomics and is related to the proteins with the increased or decreased content. However, the manuscript is addressed not only to the “proteomics audience”: the authors are also dealing with the effects  of the toxins on the behavior and they should clearly explain this term mentioning the alternating levels, upregulated and downregulated proteins. They should also mention whether they detected any post-translational modifications.

Response: Thank you for your careful and valuable advice. “ altered proteins “ means proteins at threshold as fold changes of > 1.2 or < 0.83, P < 0.05. fold changes of > 1.2 as up-regulated altered proteins, fold changes of < 0.83 as down-regulated altered proteins. No post-translational modifications had been detected in this paper, post-translational modifications need to do specific proteomics.

  1. The whole text needs attentive English editing, among most often mistakes are inconsistencies in verbs for singular and plural cases.

Response: Thank you for your careful and valuable advice. We have checked the paper and the corresponding error has been corrected.

  1. I still consider that there are two many abbreviations -at least the list of them should be available.

Response: Thank you for your careful and valuable advice. List of abbreviations had been added in lines 435-440.

  1. Line 74 -where is the reference for Sun et al?

Response: Thank you for your careful and valuable advice. The reference has been added [10]. This reference is cited twice.

  1. Fig.6. Still do not understand how in some cases (for example, membrane) the number of altered proteins can be larger than that of the identified proteins

Response: Thank you for your careful and valuable advice. Sorry for the lack of clarity, the X-axis is not the number of proteins, it's the percentage of proteins. (for example, If 10 of the altered proteins are located on the membrane, the value of X is 10/123, If 100 of the identified proteins are located on the membrane, the value of X is 100/6798 ).

  1. Line 191 should be “where” the expression content….

Response: Thank you for your careful and valuable advice. I'm a little confused, line191 is blank in my newly downloaded paper.

  1. Line 381.What does it mean “ was desalted with TFA” ? Why then it is written as an instruction “ Add TFA etc?”

Response: Thank you for your careful and valuable advice. TFA is a way of desalting from proteins, the specific steps have been revised in the paper.

  1. Line 390 should be “ fractions were collected….”

Response: Thank you for your careful and valuable advice. It has been revised in the article.

Reviewer 2 Report

Comments and Suggestions for Authors

The majority of my previous comments and suggestions were adressed. However, there are still a few outstanding issues that require clarification or response:

Firstly, regarding my suggestion to "report the results of principal component analysis or relevant tests to demonstrate the quantitative proteomics data sufficiently explains variance between treatment and control groups", the author response only stated that the analysis had been added, but upon review, the results of the PCA are still not presented in the updated manuscript. Please show or describe the PCA results to satisfy this comment.

Secondly, part of my suggestion regarding Table 1 was not fully implemented. While the title was revised, I had also proposed that "A heatmap, bar chart or other graphical representation would be more informative than a table for visualizing expression patterns of differentially expressed proteins." This second part does not appear to have been addressed. Considering the value of data visualization, please modify Table 1 or include an additional figure to improve presentation of differential protein expression patterns between groups, as suggested previously.

In the spirit of transparent and constructive peer review, I would appreciate the authors directly acknowledging and responding to each comment and suggestion provided in my initial review. Please clarify or address the outstanding issues highlighted above to fully satisfy the review. Let me know if any part requires further explanation. I aim to provide feedback that strengthens the quality and reporting of the work presented.

Comments on the Quality of English Language

Moderate editing of the English language may be required. Some inconsistent or unclear phrasing and grammatical errors affect comprehension of the content. Please proofread carefully and consider native language editing assistance if needed.

Author Response

We are thankful to the reviewer for valuable suggestions to that aided in improving our manuscript. Here, we respond to the comments in the queries and suggestions.

------

The majority of my previous comments and suggestions were adressed. However, there are still a few outstanding issues that require clarification or response:

  1. Firstly, regarding my suggestion to "report the results of principal component analysis or relevant tests to demonstrate the quantitative proteomics data sufficiently explains variance between treatment and control groups", the author response only stated that the analysis had been added, but upon review, the results of the PCA are still not presented in the updated manuscript. Please show or describe the PCA results to satisfy this comment.

Response: Thank you for your careful and valuable advice. “Figure 4. Correlation of the protein intensity among four biological replicates of control and shellfish poisoning model groups” had been added.

  1. Secondly, part of my suggestion regarding Table 1 was not fully implemented. While the title was revised, I had also proposed that "A heatmap, bar chart or other graphical representation would be more informative than a table for visualizing expression patterns of differentially expressed proteins." This second part does not appear to have been addressed. Considering the value of data visualization, please modify Table 1 or include an additional figure to improve presentation of differential protein expression patterns between groups, as suggested previously.

Response: Thank you for your careful and valuable advice. The differentially expressed protein,which discuss in this article are shown in Table 1. As you suggested, I have added Supplementary Materials table s1, which includes all differentially expressed proteins.

Reviewer 3 Report

Comments and Suggestions for Authors

Dear Editor, Greetings

I report that I have read in detail the new version of this article, and I consider that the authors still maintain important deficiencies in the way of presenting and discussing their data.

 The article is interesting, but it lacks important experimental and analytical controls that would give rigor to what they propose. Likewise, the introduction is confusing and poorly addressed by the authors, who do not know or confuse some basic topics related to the effects of PST.

I make some comments, but this needs to be revised in detail, also considering that the authors do not respond adequately to the concerns raised in the previous review.

Comments:

Figures: revise the figures included.

Line 22-23: I can't understand this sentence.

Line 26: correct your punctuation.

Line 28-30: correct the names of the toxin groups.

Line 42: What is the evidence that the toxins are cardiotoxic?

Line 43: What do you mean by high?

Line 70-80: include the objective and hypothesis of the study.

Line 84: What is a model group?

Line 84-85: What is the difference between acute and chronic exposure?

Comments on the Quality of English Language

Extensive editing of English language required

Author Response

We are thankful to the reviewer for valuable suggestions to that aided in improving our manuscript. Here, we respond to the comments in the queries and suggestions.

------

Dear Editor, Greetings

I report that I have read in detail the new version of this article, and I consider that the authors still maintain important deficiencies in the way of presenting and discussing their data.

 The article is interesting, but it lacks important experimental and analytical controls that would give rigor to what they propose. Likewise, the introduction is confusing and poorly addressed by the authors, who do not know or confuse some basic topics related to the effects of PST.

I make some comments, but this needs to be revised in detail, also considering that the authors do not respond adequately to the concerns raised in the previous review.

Comments:

  1. Figures: revise the figures included.

Response: Thank you for your careful and valuable advice. I'm a little confused, if you mentioned the number of figures, I've modified it to 6.

  1. Line 22-23: I can't understand this sentence.

Response: Thank you for your careful and valuable advice. The sentence had been modified.

  1. Line 26: correct your punctuation.

Response: Thank you for your careful and valuable advice.The punctuation had been modified.

  1. Line 28-30: correct the names of the toxin groups.

Response: Thank you for your careful and valuable advice. we had deleted “azaspir acid (AZA)”.

  1. Line 42: What is the evidence that the toxins are cardiotoxic?

Response: Thank you for your careful and valuable advice. “The toxins are cardiotoxic” had reported in “Evolutional Researches about Detection of Paralytic Shellfish Toxins, Forensic Science and technology, 2023,48(4):418-425.” but we didn't find any other literature, so we deleted “and cardiovascular system”.

  1. Line 43: What do you mean by high?

Response: Thank you for your careful and valuable advice. The “high” means “reach up to” , to make it easier to read, we change it to “After eating aquatic products with PSTs content as high as 80 μg / 100 g”.

  1. Line 70-80: include the objective and hypothesis of the study.

Response: Thank you for your careful and valuable advice. This paragraph has been revised.

  1. Line 84: What is a model group?

Response: Thank you for your careful and valuable advice. The model group means shellfish poisoning nodel group.

  1. Line 84-85: What is the difference between acute and chronic exposure?

Response: Thank you for your careful and valuable advice. The mice LD50 of saxitoxin is orally 382μg/kg. The dosage (5μg/kg) we determined was lower than LD50, was not lethal to mice, and was given for a long time, so it was chronic exposure.

Reviewer 4 Report

Comments and Suggestions for Authors

Unfortunatelly, my revomendation was not implemented by authors. Authorths added the same images with more bigger magnificaltion - but does not enought to undestend does  “granulovacuolar degeneration” exisn or not. Pay attantion that both control and model imgases have “granulovacuolar degeneration” cells. I strongly recommend (1) either remove the hystolologicall data from the manuscript, (2) or prove more adequit microscopic techniques to prove “granulovacuolar degeneration”. 

Author Response

We are thankful to the reviewer for valuable suggestions to that aided in improving our manuscript. Here, we respond to the comments in the queries and suggestions.

------

Unfortunatelly, my revomendation was not implemented by authors. Authorths added the same images with more bigger magnificaltion - but does not enought to undestend does  “granulovacuolar degeneration” exisn or not. Pay attantion that both control and model imgases have “granulovacuolar degeneration” cells. I strongly recommend (1) either remove the hystolologicall data from the manuscript, (2) or prove more adequit microscopic techniques to prove “granulovacuolar degeneration”.

Response: Thank you for your careful and valuable advice. According to your suggestion, the hystolologicall data has been deleted.

Round 3

Reviewer 1 Report

Comments and Suggestions for Authors

no new comments

Comments on the Quality of English Language

I still think that moderate editing of English is required

Author Response

Response: Thank you for your careful and valuable advice.

Reviewer 2 Report

Comments and Suggestions for Authors

The authors have diligently addressed the majority of the required comments and made appropriate changes to the manuscript, resulting in an improved overall quality.

Comments on the Quality of English Language

Throughout the article, there are instances where sentence fragments are used. To enhance clarity and readability, it would be beneficial to restructure these fragments into complete sentences.

For instance, in Line 85, the original fragment states: "As shown in Figure 1, the weight of mice in the model group was significantly decreased from that in the control group during the second week to the fifth week."

A revised sentence could be: "Figure 1 demonstrates a significant decrease in the weight of mice in the model group compared to the control group from the second week to the fifth week."

Similarly, in Line 86, the original fragment is: "As shown in Figure 1, the weight of mice in the model group was significantly decreased from that in the control group during the second week to the fifth week."

The revised sentence could be: "Figure 1 illustrates a significant decrease in the weight of mice in the model group compared to the control group from the second week to the fifth week."

Throughout the article, there are instances where sentence fragments are used. It is important to note that the examples provided are just a few illustrations, and there may be additional abnormal phrasing in the text that I have not listed.

Overall, the English used in this article is generally clear and understandable. However, there are areas where improvements could be made to enhance readability and clarity.

Author Response

We are thankful to the reviewer for valuable suggestions to that aided in improving our manuscript. Here, we respond to the comments in the queries and suggestions.

------

Throughout the article, there are instances where sentence fragments are used. To enhance clarity and readability, it would be beneficial to restructure these fragments into complete sentences.

For instance, in Line 85, the original fragment states: "As shown in Figure 1, the weight of mice in the model group was significantly decreased from that in the control group during the second week to the fifth week."

A revised sentence could be: "Figure 1 demonstrates a significant decrease in the weight of mice in the model group compared to the control group from the second week to the fifth week."

Similarly, in Line 86, the original fragment is: "As shown in Figure 1, the weight of mice in the model group was significantly decreased from that in the control group during the second week to the fifth week."

The revised sentence could be: "Figure 1 illustrates a significant decrease in the weight of mice in the model group compared to the control group from the second week to the fifth week."

Throughout the article, there are instances where sentence fragments are used. It is important to note that the examples provided are just a few illustrations, and there may be additional abnormal phrasing in the text that I have not listed.

Overall, the English used in this article is generally clear and understandable. However, there are areas where improvements could be made to enhance readability and clarity.

Response: Thank you for your careful and valuable advice. According to your suggestion, we have revised and polished the full text.

Reviewer 3 Report

Comments and Suggestions for Authors

Dear Editor Greetings.

I have read in detail the revised version of the article, as well as the answers to the questions posed to the authors.

I consider that this version has changed remarkably and has incorporated the corrections to eliminate the doubts raised.

I suggest a revision of the text for some minor errors in editing and language.

Comments on the Quality of English Language

Moderate editing of English language required.

Author Response

We are thankful to the reviewer for valuable suggestions to that aided in improving our manuscript. Here, we respond to the comments in the queries and suggestions.

------

Dear Editor Greetings.

I have read in detail the revised version of the article, as well as the answers to the questions posed to the authors.

I consider that this version has changed remarkably and has incorporated the corrections to eliminate the doubts raised.

I suggest a revision of the text for some minor errors in editing and language.

Response: Thank you for your careful and valuable advice. According to your suggestion, we have revised and polished the full text.

Reviewer 4 Report

Comments and Suggestions for Authors

Authors implemented all of my recomendations. I think the MS can be published in Toxins journal.

Author Response

(The authors gave the same response as above.)
